# Cardiac Magnetic Resonance Strain in Beta Thalassemia Major Correlates with Cardiac Iron Overload

**DOI:** 10.3390/children10020271

**Published:** 2023-01-31

**Authors:** Deidra Ansah, Nazia Husain, Alexander Ruh, Haben Berhane, Anthony Smith, Alexis Thompson, Andrew De Freitas, Cynthia K. Rigsby, Joshua D. Robinson

**Affiliations:** 1Department of Pediatrics, Texas Children’s Hospital at Baylor College of Medicine, Houston, TX 77030, USA; 2Department of Pediatrics, Ann & Robert H. Lurie’s Children’s Hospital of Chicago, Northwestern University Feinberg School of Medicine, Chicago, IL 60611, USA; 3Department of Radiology, Northwestern University Feinberg School of Medicine, Chicago, IL 60611, USA; 4Department of Medical Imaging, Ann & Robert H. Lurie’s Children’s Hospital of Chicago, Northwestern University Feinberg School of Medicine, Chicago, IL 60611, USA; 5Department of Pediatrics, Children’s Hospital of Philadelphia, Perelman School of Medicine at the University of Pennsylvania, Philadelphia, PA 19104, USA

**Keywords:** strain, beta thalassemia major, Iron Overload, Iron metabolism, cardiomyopathies, magnetic resonance imaging, Parametric mapping, T2*

## Abstract

Background: Beta thalassemia major (Beta-TM) is an inherited condition which presents at around two years of life. Patients with Beta-;TM may develop cardiac iron toxicity secondary to transfusion dependence. Cardiovascular magnetic resonance (CMR) T2*, a technique designed to quantify myocardial iron deposition, is a driving component of disease management. A decreased T2* value represents increasing cardiac iron overload. The clinical manifestation is a decline in ejection fraction (EF). However, there may be early subclinical changes in cardiac function that are not detected by changes in EF. CMR-derived strain assesses myocardial dysfunction prior to decline in EF. Our primary aim was to assess the correlation between CMR strain and T2* in the Beta-TM population. Methods: Circumferential and longitudinal strain was analyzed. Pearson’s correlation was calculated for T2* values and strain in the Beta-TM population. Results: We identified 49 patients and 18 controls. Patients with severe disease (low T2*) were found to have decreased global circumferential strain (GCS) in comparison to other T2* groups. A correlation was identified between GCS and T2* (r = 0.5; *p* < 0.01). Conclusion: CMR-derived strain can be a clinically useful tool to predict early myocardial dysfunction in Beta-TM.

## 1. Introduction

Thalassemias are a group of inherited blood disorders characterized by abnormal hemoglobin production due to genetic mutations in the alpha or beta globin genes. One of the most common and most severe subtypes is beta thalassemia major (Beta-TM) [1]. In its most severe form, Beta-TM results in hemolysis, ineffective erythropoiesis, and severe anemia requiring chronic blood transfusions. Many patients develop iron overload which can result in cardiac and liver toxicity. Heart failure secondary to cardiac siderosis is the leading cause of death in this patient population [2,3]. Since the late 1980s, iron chelation has become the standard of care to prevent the morbidity and mortality of the associated iron overload [2]. Measurement of T2* relaxation time by cardiovascular magnetic resonance (CMR) was introduced in the late 1990s to allow for assessment of myocardial iron load. The use of T2* imaging has been a driving component of iron chelator dose management and has led to a significant reduction of the mortality associated with iron overload-related cardiomyopathy in the Beta-TM population [2,4,5]. The literature has demonstrated that T2* values less than 10 ms are associated with an increase in heart failure and ventricular arrhythmia [6]. Normal T2* values are generally reported as greater than 20 ms, while a mean myocardial T2* around 40 ms when measured at the interventricular septum has also been reported [5,7]. With increasing iron overload in the myocardium, there may be early subclinical changes in cardiac function not detected by changes in ejection fraction (EF) [8]. The etiology and pathogenesis of cardiomyopathy seen in the Beta-TM population is multifactorial. Cardiac siderosis usually presents as a restrictive cardiomyopathy, although a dilated cardiomyopathy presentation secondary to high output failure with chronic anemia is also seen in these patients [1]. Although T2* imaging provides a precise assessment of myocardial iron load, patients with the same iron load can present with varying degrees of cardiomyopathy [2]. This complex pathophysiologic interplay suggests a need for multi-faceted characterization of the myocardium in this patient population.

CMR allows for accurate and comprehensive assessment of ventricular systolic function in addition to the evaluation of myocardial iron load. Strain describes the change in length in a particular orientation, and when applied to the heart it represents myocardial deformation from a relaxed to contracted state; it is dimensionless and expressed in %. Peak contractility is represented by a more negative value due to myocardial contractility being a shortening mechanism. Strain rate is the amount of deformation as a function of time expressed as s^−1^ [9]. CMR strain imaging has been shown to detect early contractile dysfunction prior to decline in EF [10] and offers an opportunity to further characterize the myocardium in patients with Beta-TM. There is scant literature evaluating the clinical utility of CMR strain imaging within the Beta-TM population and there is great need for understanding the underlying mechanics in cardiac dysfunction in this patient population. The primary aim of our study was to investigate the relationship between feature tracking CMR-derived strain and T2* in efforts to add to the limited existing data. We hypothesized that declining CMR strain correlates with decreasing T2* values in the Beta-TM population which may play a significant role in the detection of pre-clinical cardiac dysfunction in this patient population.

## 2. Materials and Methods

This retrospective study was approved by our Institutional Review Board (IRB) and is Health Insurance Portability and Accountability Act (HIPAA) compliant. The informed consent necessity was waived by the IRB. The controls provided written consent for the general use of their data for investigations within our imaging department; they were not recruited for the purposes of this study. The control data was retrospectively used in our investigation under a waiver from the IRB.

Subjects: The study population consisted of patients with a diagnosis of a thalassemia syndrome including transfusion-dependent beta thalassemia major and hemoglobin E beta thalassemia. The subjects were identified through an institutional database consisting of CMR T2* data from 2010 to 2018. Beta-TM is an inherited hemoglobinopathy with pediatric onset. However, with regular blood transfusions and chelation therapy, the onset of heart failure in Beta-TM is more commonly seen in the third decade of life and beyond; thus, our cohort was older than 18 years of age [1]. Inclusion criteria: patients with a CMR study performed after 2010 including T2* sequence and post-processing, a CMR EF greater than 50%, and no known history of non-iron overload-related cardiomyopathy. The total cohort was divided into four sub-groups according to T2* values. These groups included T2* < 10 ms, T2* 10–20 ms, T2* 20–30 ms, and T2* greater than 30 ms. Although T2* values > 20 ms are considered normal, because the mean T2* in the general population has been reported around 40 ms, we felt that the T2* 20–30 ms sub-group was a valuable cohort to include for the full spectrum of disease burden. The controls were age-matched healthy volunteers recruited from the community between the years of 2017 and 2018 using flyer-based advertisement. The exclusion criteria for the controls consisted of any history of cardiovascular disease, pregnancy, and a body mass index greater than 36 kg/m^2^.

Data Collection: Demographic data was collected using electronic medical records at our institution. The demographic data collected included: age, gender, and date of study. The hematology data collected included: thalassemia genotype, pre-transfusion hemoglobin, transfusion volume for the year preceding the CMR study, ferritin, chelator therapy, and splenectomy history.

Cardiovascular Magnetic Resonance: CMR was performed on a 1.5 T magnetic resonance imaging (MRI) system (MAGNETOM Aera, Siemens Healthineers, Erlangen, Germany) for both subjects and controls. After scout imaging, functional assessment was performed using retrospectively gated cine balanced steady state free precession (bSSFP) images in standard long axis geometries (two-, three-, and four-chamber view) as well as in short axis orientation with full ventricular coverage from base to apex. Although scanned on the same MRI system, the controls were scanned at a different facility using different parameters.

Post-Processing: Short axis slices were analyzed to calculate left ventricular volumes and EF using QMass (version 7.2, Medis Medical Imaging Systems, Leiden, The Netherlands). T2* post-processing was performed for clinical purposes using a customized MATLAB tool (MathWorks, Natick, MA, USA). T2* sequences were post-processed at the time the study was performed according to our institution’s previously published standardized method [11]. A region of interest (ROI) was contoured within the interventricular septum of the three ventricular short axis slices. For each ROI, the mean intensity was evaluated and plotted as a function of echo time (TE). T2* was then calculated by fitting the intensity data to a monoexponential decay using a nonlinear Levenberg–Marquardt algorithm. These values were subsequently included in the CMR report and in our institutional database. Our institution transitioned to use of Medis suite MR T2/T2* analysis (version 3.1; Medis Medical Imaging Systems, Leiden, The Netherlands) for post-processing T2* in 2018, affecting one patient in our population. The post-processing technique described above remained the same despite the change in software.

CMR strain analysis was performed by two blinded observers using the feature tracking software QStrain (version 3.1; Medis Medical Imaging Systems, Leiden, The Netherlands). Longitudinal and circumferential strain was assessed by tracing endocardial borders on cine bSSFP images. The longitudinal strain tracing was performed in the two-, three-, and four-chamber views and for circumferential strain in short axis views at the base, mid, and apex (Figure 1). The endocardial borders were manually drawn in end-diastole and end-systole. An American Heart Association (AHA) 16-segment left ventricular (LV) model [12] was used for creating a map of LV longitudinal and circumferential segmental strain and strain rate. Peak strain values and peak systolic and diastolic strain rates were detected and averaged over all LV segments to obtain global peak values.

Statistical Analysis: Statistical analysis was performed using MATLAB (MathWorks, Natick, MA). Strain values were compared within the T2* groups using a four-group comparison and between controls and individual patient groups using pairwise comparisons. A Bonferroni correction was applied where multiple pairwise comparisons were performed; the significance level in the post-hoc analysis was 0.013. Normal distribution was tested using a Lilliefors test. For normally distributed data, an analysis of variance (one-way ANOVA) was used and for non-normally distributed data a Kruskal–Wallis test was used. For subsequent pairwise comparisons, a t-test was used for normally distributed data and a Wilcoxon rank-sum test was used for non-normally distributed data. The Pearson’s correlation was calculated for the T2* values and strain, T2* values and strain rate, and strain and hematological variables including transfusion volume, ferritin, and hemoglobin. Bland–Altman plots were used to assess inter-observer agreement for circumferential and longitudinal strain. *p* value < 0.05 was considered significant (in the absence of multiple pairwise comparisons). A receiver operating characteristic (ROC) curve was utilized to define the sensitivity and specificity of global circumferential strain (GCS) and diastolic circumferential strain rate values associated with a T2* less than 10 ms. A strain and strain rate cut-off were performed as two-sided at a significance level of *p* < 0.05.

## 3. Results

### 3.1. Study Population

There was a total of 49 patients and 18 controls. Of the 49 patients, 32 were female and 17 were male. In the control group, 8 were female and 10 were male (*p* = 0.02). Table 1 represents the demographic and clinical characteristics of the controls and the Beta-TM patients categorized according to their T2* values. A total of 22 subjects (49%) had a T2* value greater than 30 ms, 8 patients (16%) had a T2* value of 21–30 ms, 10 patients (20%) had a T2* of 11–20 ms, and 9 patients (18%) had a T2* of less than 10 ms. There were no significant differences in age, body surface area, or EF between the Beta-TM patients and the controls. There was no difference in pre-transfusion hemoglobin or transfusion volume for any of the T2* groups. An increased ferritin level was noted in the T2* group 10 ms (*p* < 0.01).

### 3.2. Myocardial Functional Analysis

The global strain and strain rate results of the T2* groups are shown in Table 2. There was a difference amongst the groups when evaluating GCS (*p* < 0.01) with the T2* <10 ms group having the lowest strain value (suggesting less myocardial deformation) of −26.8 ± 2.7%.

Pairwise comparisons demonstrated a reduced GCS when comparing the T2* < 10 ms group to all the other groups (Figure 2).

The global diastolic circumferential strain rate was also reduced in the T2* < 10 ms group at 1.7 ± 0.4 s^−1^_._ In pairwise comparisons, the global diastolic circumferential strain rate was significantly different when comparing T2* < 10 ms group to the T2* 21–30 ms and T2* > 30 ms groups (Figure 3).

The comparison of the Beta-TM patients to the controls is presented in Table 3. The controls had a higher GCS (−31.4% ± 3.9) and global longitudinal strain (GLS) (−26.2% ± 3.6) than the T2* < 10 ms group (GCS: −26.8 ± 2.7; GLS: −22.8 ± 2.7) with *p* < 0.01 and 0.04, respectively. The controls had lower circumferential and longitudinal strain rates when compared to the T2* > 30 ms group.

Figure 4 and Figure 5 represent the segmental circumferential strain and diastolic circumferential strain rate values of patients with a T2* < 10 ms and >30 ms compared to controls. Although there were some observed segmental differences between the patient group and the controls, there were no distinct trends.

Significant correlations were identified between T2* and GCS (r = 0.5, *p* <0.01) and between T2* and peak diastolic circumferential strain rate (r = 0.4, *p* <0.01) as shown in Figure 6. There was no significant correlation between peak systolic circumferential strain rate (r = 0.21, *p* = 0.14), longitudinal strain (r = 0.25, *p* = 0.83), or longitudinal strain rate (systolic: r = 0.22, *p* = 0.88; diastolic: r = 0.23, *p* = 0.12) and T2* values.

Analysis of the ROC curve (Figure 7) showed an area under the curve of 0.89 (95% CI 0.722–0.973) for GCS with the best threshold value to determine a T2* < 10 ms of −29.97% with a sensitivity of 78% and a specificity of 95%. The area under the curve for diastolic circumferential strain rate was 0.84 (95% CI 0.628–0.946) with the best threshold value for determining T2* < 10 ms of 1.96 s^−1^ with a sensitivity of 67% and a specificity of 92%. Bland–Altman analysis for GCS for the patient population is shown in Figure 8 and demonstrates inter-observer agreement.

### 3.3. Hematological Data

Correlations between the hematological variables (hemoglobin, ferritin and transfusion volume) and our T2* groups were assessed. A statistically significant negative correlation was identified between transfusion volume and longitudinal strain (r = −0.76; *p* = 0.03) as well as between transfusion volume and longitudinal strain rate (systole: r = −0.72; *p* = 0.04; diastole: r = −0.89; *p* = <0.01) This difference was found only in the T2* group 11–20 ms. A significant negative correlation was identified between hemoglobin and diastolic longitudinal strain rate in the T2* > 30 ms group (r = −0.50; *p* = 0.04) as well as between ferritin and GCS (r = −0.38; *p* = 0.01) for the entire cohort of patients. These findings are summarized in Table 4.

## 4. Discussion

Using a feature tracking method to acquire CMR strain data, our center demonstrated a decline in CMR-derived strain in Beta-TM patients with iron overload. Our primary observation was that in the setting of similar ejection fractions amongst T2* values, there was a decrease in GCS in patients with abnormal T2* values in comparison to other Beta-TM patients with higher T2* values and in comparison to controls. We identified a correlation between T2* values and GCS, as well as a correlation between T2* values and circumferential diastolic strain rate. The thresholds derived from the ROC curve analyses were GCS of −29.97% and global diastolic circumferential strain rate of 1.96 s^−1^, showing good performance in the differentiation of the cases with the most severe disease state (T2* < 10 ms) with a sensitivity of 78% and a specificity of 95% for the GCS and a sensitivity of 67% and a specificity of 92% for the diastolic circumferential strain rate.

Studies evaluating the relationship between myocardial deformation and T2* values have primarily focused on echocardiography-derived strain. In a prospective evaluation of 55 patients with Beta-TM, Di Odoardo et al. found that in comparing strain analysis of patients with Beta-TM to healthy controls, Beta-TM patients showed a significant decrease in radial and circumferential strain indices. They did not find significant differences between strain indices in Beta-TM patients who had low T2* and Beta-TM patients who had normal T2* values. They additionally found no significant correlation between T2* values and echocardiographic strain [13]. Ari et al. compared echocardiographic strain indices of 30 patients with Beta-TM to 30 healthy controls. They also compared the strain indices of Beta-TM patients with normal T2* values to those with abnormal values. Similar to Di Odoardo et al., they found a decrease in strain of Beta-TM patients in comparison to their healthy counterparts. In contrast to the Di Odoardo findings, when comparing varying cohorts of T2* groups they found that strain indices in patients with pathological T2* values (<20 ms) were significantly lower than the patients with normal T2* values [14].

There are few studies evaluating the use of magnetic resonance strain and its relationship to T2*. Rezaeian et al. described their evaluation of cardiac strain within the Beta-TM population also using a feature tracking method. They evaluated global circumferential, global longitudinal, and global radial strain in both ventricles. Their group identified significant correlations between cardiac T2* values and all strain parameters in addition to a significant difference between all left ventricular strain values between Beta-TM patients with myocardial iron overload in comparison to their healthy control group [15]. Conversely, in a prospective evaluation of the relationship between feature tracking MR-derived global longitudinal, radial, and circumferential strain and T2*, Ojha et al. identified a correlation only between global radial strain of Beta-TM patients and T2*. There was no significant difference between the global radial, circumferential, or longitudinal strain values amongst their patient population with and without T2* evidence of myocardial iron overload [16].

We did not find a significant difference amongst our cardiac T2* groups when evaluating global or segmental longitudinal strain. Our negative finding in reference to longitudinal strain may be secondary to some of the inherent pitfalls with feature tracking at the basal segments. This can occur due to the thinner walls, particularly at the hinge points of the mitral valve annulus [10] and/or due to the possible extrinsic sternal compression on basal segments which is more enhanced in individuals with anterior chest wall deformity, such as those with pectus excavatum [17]. Additionally, the iron deposition pattern in iron overload cardiomyopathy has been described as occurring primarily at the subepicardial fibers. The effects on longitudinal strain may be a later finding in iron overload cardiomyopathy because of the longitudinal fiber distribution pattern being predominantly endocardial [18,19]. However, the varying findings amongst our investigation and those of Rezaeian et al. and Ojha et al., alludes to a multifactorial nidus of the cardiac dysfunction seen in this patient population and suggests that the deposition pattern of iron overload cardiomyopathy continues to require further investigation. We did not evaluate radial strain as the literature has demonstrated that CMR-derived longitudinal and circumferential strain demonstrate greater reliability than radial strain [20,21,22]. In line with the literature, our data demonstrated inter-observer agreement with circumferential and longitudinal strain amongst our patient population.

Similar to GCS, the T2* group less than 10 ms had lower global circumferential diastolic strain rate in comparison to the other T2* groups. There are varying reports in the literature regarding the prevalence of diastolic dysfunction in this patient population and how it relates to T2* values. Mancuso et al. recently reviewed their evaluation of heart failure in Beta-TM and found diastolic failure of the left ventricle to be a significant contributor to cardiac dysfunction [23]. Similarly, Efthimiadis et al. in their 15-year follow up of 45 patients with Beta-TM found a significant association between cardiac-related mortality and restrictive left ventricular physiology. They additionally identified an increase in left ventricular diastolic dysfunction in patients who were less compliant with their chelation therapy [24]. The sequence of cardiac dysfunction in Beta-TM has been described as iron deposition leading to diastolic dysfunction that likely precedes systolic dysfunction [25]. Our finding of diastolic strain rate being uniformly lower in Beta-TM patients with lower T2* values may be reflective of the known early changes in diastology seen in this patient population [24,25,26]. The ejection fractions amongst the T2* groups were not significantly different, which corroborates established data that even with known differences in myocardial iron overload, a change in EF is usually a late finding in this patient population group [2,4,14,19].

Higher imaging temporal resolution is associated with higher longitudinal and circumferential strain and strain rate values [27]. The Beta-TM group had slightly higher temporal resolution in comparison to the controls. Despite the association with temporal resolution and strain, our results maintain a demonstration of lower GCS when comparing T2* patients with significant myocardial iron overload (T2* < 10 ms) to controls. The Beta-TM T2* group > 30 ms had a higher GCS than the control group. Given the high stroke volume and high cardiac output state that characterizes Beta-TM prior to clinical signs of iron overload cardiomyopathy [28], higher strain values in less severe disease states likely reflects early hyperdynamic function. Additionally, Table 3 demonstrates several instances of higher strain rates in the Beta-TM group throughout the range of T2* values in comparison to controls. This finding was also described by Bay et al. [29] and may represent a compensatory mechanism as previously mentioned; however, there is no clear explanation for this observation in our patients with severe disease states. Although regional cardiac dysfunction secondary to non-homogenous deposition of iron has been well described [29,30], our segmental analysis of circumferential strain and diastolic circumferential strain rate in comparison to controls did not demonstrate any clear trends.

Ferritin serves as an indirect marker of iron stores in the body and is increased in the transfusion-dependent population [1]. The literature has consistently demonstrated a poor correlation between serum ferritin and cardiac T2* values [5,31,32,33] and therefore our group did not assess the correlation of ferritin to T2* values. We did identify a significant difference amongst the ferritin values of the T2* groups. Our group sought to evaluate the relationship between CMR-derived strain and other clinically useful hematological variables. We identified a negative correlation between GCS and ferritin for the entire T2* cohort. Although higher systemic iron stores do not correlate with myocardial iron stores, our data suggests that elevated systemic iron stores are associated with a decline in myocardial function. There is likely benefit in future evaluation of the combined diagnostic utility of these parameters. There was no consistent trend when assessing the correlation between strain, transfusion volume, and hemoglobin. This suggests that the inconsistent significant findings may not be clinically relevant.

The limitations of our study include its retrospective nature as well as the small number of patients within our T2* sub-groups. The normal values for CMR-strain using feature tracking methods were variable. Abnormal values for LV GCS have been reported as greater than −17.4% to −27.2% [20,21,34,35]. With a larger cohort, perhaps our results may have demonstrated a more marked difference in strain amongst our varying T2* groups and when comparing our T2* groups to controls.

In conclusion, CMR-derived GCS was reduced in Beta-TM patients with normal EF and T2* < 10 ms compared to controls. Decreasing T2* values correlated with decreased GCS and diastolic strain rate in this population. CMR strain may be used in combination with T2* to predict early LV dysfunction in Beta-TM. The routine addition of strain assessment to Beta-TM CMR scans allows for a comprehensive structure and function assessment for management strategies in this patient population.

## Figures and Tables

**Figure 1 children-10-00271-f001:**
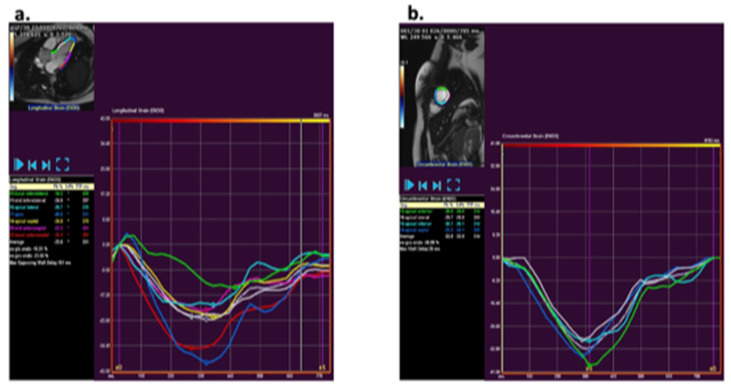
Analysis of strain using QStrain (version 3.1; Medis, Leiden, The Netherlands Medis) via a feature tracking method. (**a**) Example of longitudinal strain post-processing at the three-chamber slice. (**b**) Example of circumferential strain post-processing at mid-ventricular short axis apical slice.

**Figure 2 children-10-00271-f002:**
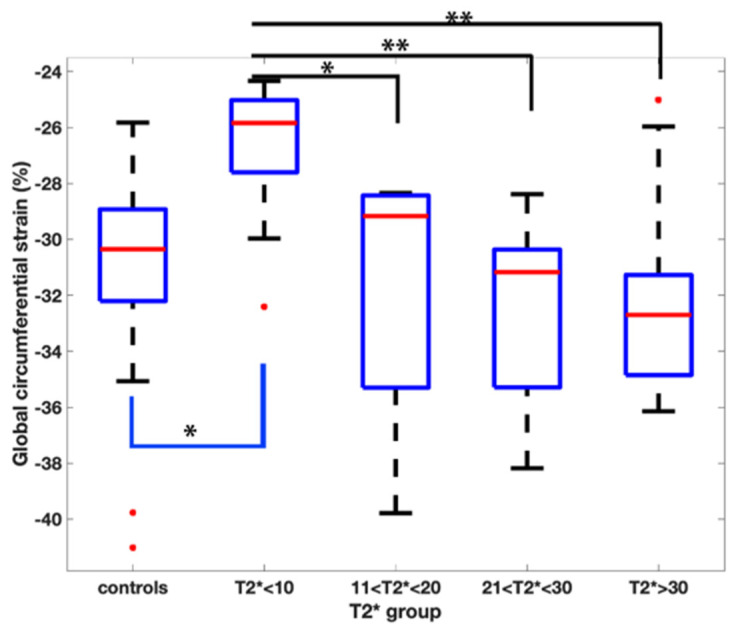
Pairwise comparison of the global circumferential strain values amongst the T2* groups. Asterix represents statistical significance with a *p* value of less than 0.01 with a Bonferroni correction applied.

**Figure 3 children-10-00271-f003:**
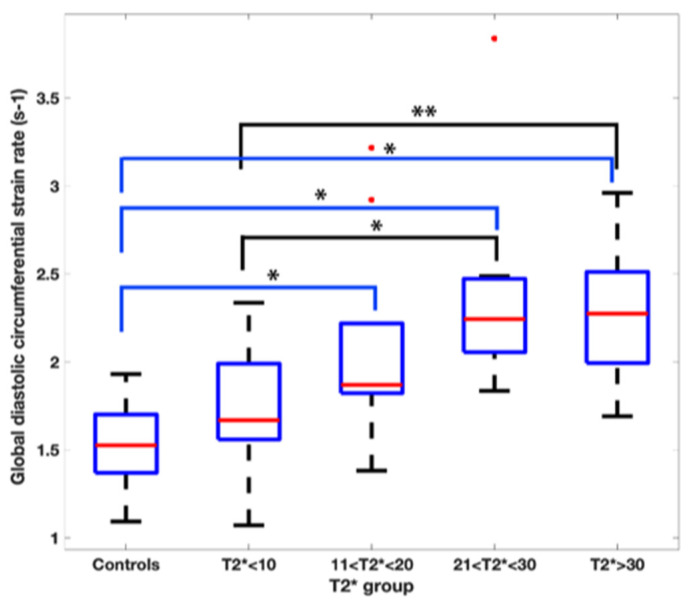
Pairwise comparison of the global diastolic circumferential strain rate values amongst the T2* groups. Asterix represents statistical significance with a *p* value of less than 0.01 with a Bonferroni correction applied.

**Figure 4 children-10-00271-f004:**
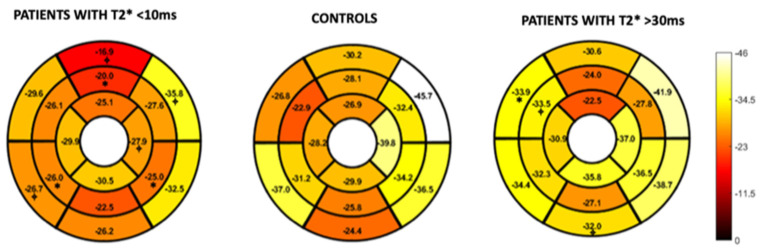
16-segment model displaying segmental analysis of endocardial circumferential strain (%). Patients with T2* less than 10 ms and greater than 30 ms are compared to healthy controls. *; *p* < 0.05 ⌖; *p* value < 0.01.

**Figure 5 children-10-00271-f005:**
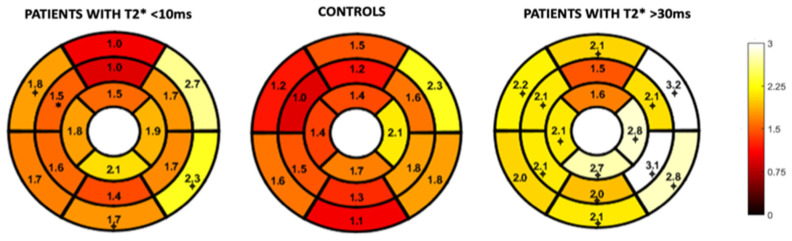
16-segment model displaying segmental analysis of endocardial circumferential diastolic strain rate (s^−1^). Patients with T2* less than 10 ms and greater than 30 ms are compared to healthy controls *; *p* < 0.05 ⌖; *p* value < 0.01.

**Figure 6 children-10-00271-f006:**
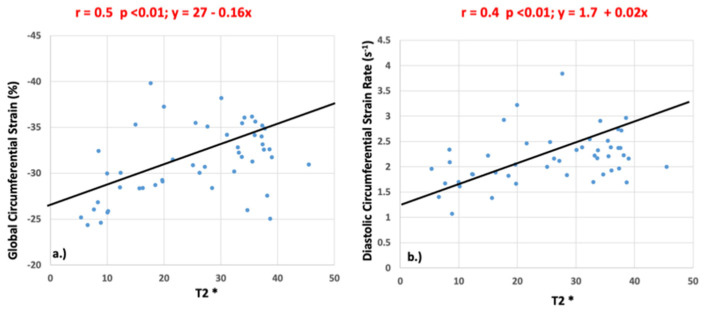
Pearson’s correlation values for global circumferential strain (**a**) and diastolic circumferential strain rate (**b**) when compared to T2*.

**Figure 7 children-10-00271-f007:**
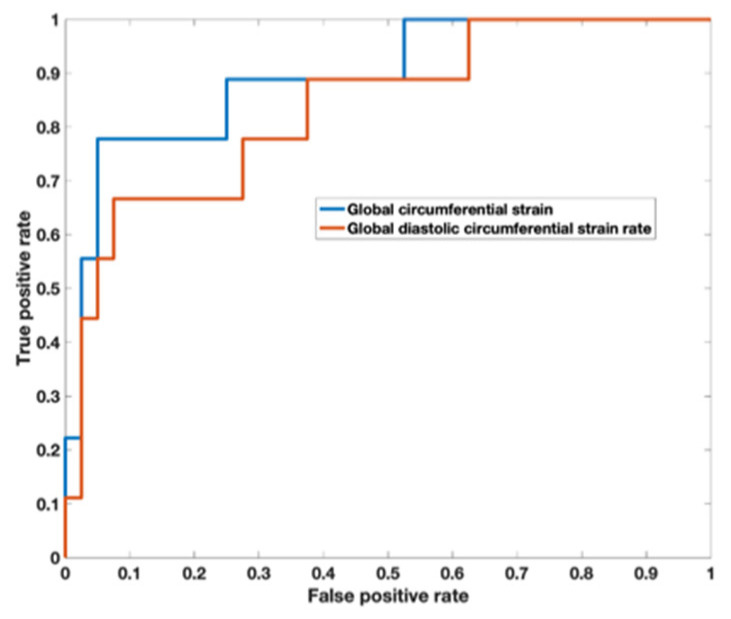
Receiver operator characteristic curve demonstrates an area under the curve of 0.89 (95% CI 0.722–0.973) for global circumferential strain with the best threshold value to determine a T2* < 10 ms of −29.97%. The area under the curve for diastolic circumferential strain rate was 0.836 (95% CI 0.628–0.946) with the best threshold value for determining T2* < 10 ms of 1.96 s^−1^.

**Figure 8 children-10-00271-f008:**
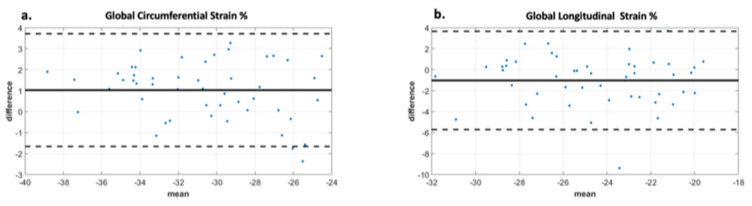
Bland Altman analysis demonstrating interobserver agreement for two observers. Image (**a**) demonstrates excellent agreement between observers for global circumferential strain with a bias of 1. The limits of agreement are +/− 2.8 which is less than 10% of the mean value. Image (**b**) demonstrates moderate agreement between observers for global longitudinal strain with bias -1 and limits of agreement +/− 4.8, which is less than 20% of the mean value. The displayed limits of agreement are two standard deviations.

**Table 1 children-10-00271-t001:** Demographic and hematological data of the T2* groups and controls.

	T2* < 10 ms (n = 9)	T2* 11–20 ms (n = 10)	T2* 21–30 ms (n = 8)	T2* > 30 ms (n = 22)	All pts (n = 49)	Controls (n = 18)	*p* Value
Age (yr)	27 ± 4.6	30 ± 6.9	26 ± 0	30 ± 12.2	29 ±9.6	30 ± 7.8	0.04
Gender	m:5 f:4	m:6 f:4	m:0 f:8	m:6 f:16	m:17 f:32	m:10 f:8	**0.02**
Weight (kg)	62 ± 20	63 ± 10	56 ± 7	59 ± 10	60 ± 12	67 ± 3	**0.01**
BSA (m^2^)	1.7 ± 0.3	1.7 ± 0.2	1.5 ± 0.1	1.6 ± 0.1	1.6 ± 0.2	1.8 ± 0.1	0.21
Ejection fraction (%)	55 ± 4	56 ± 6	57 ± 4	58 ± 4	57 ± 5	65 ± 0.7	0.06
Pre-transfusion Hgb for year of MR (g/dL)	10.0 ± 1.2	9.5 ± 1.3	8.8 ± 0.9	9.7 ± 1.0	9.5 ± 1.1		0.1
Transfusion volume for year of MRI (ml/kg/year)	111.0 ± 53.0	87.8 ± 69.5	101.7 ± 93.9	111.0 ± 101.7	104 ± 79.8		0.9
Ferritin (µg/L)	6006 ± 4412	1903 ± 1221	2127 ± 1296	1310 ± 834	2678 ± 2817		**<0.01**
Cardiac T2* (ms)	8.4 ± 1.6	16.7 ± 2.9	26.5 ± 2.6	36.2 ± 3.1	19.1 ± 8.0		**<0.01**
Presence of a spleen (n)	3 (33%)	4 (40%)	4 (50%)	8 (36%)	19(39%)		0.9
Chelator Medication (n)							
Deferasirox	6 (66%)	7 (70%)	5 (63%)	14 (64%)	32 (65%)		0.39
Desferal	4 (44%)	3 (30%)	1 (13%)	2 (9%)	10 (20%)		0.25
Deferiprone	3 (33%)	2 (20%)	2 (25%)	3 (14%)	10 (20%)		0.67

Continuous variables represented as mean ± standard deviation; bolded values indicate statistically significant difference (*p*-value < 0.05). T2*: T2 star imaging.

**Table 2 children-10-00271-t002:** Myocardial deformation data of the T2* groups.

		T2* < 10 ms (n = 9)	T2* 11–20 ms (n = 10)	T2* 21–30 ms (n = 8)	T2* > 30 ms (n = 22)	*p* Value
**Global circumferential strain (%)**		−26.8 ± 2.7	−31.5 ± 4.3	−32.5 ± 3.3	−32.4 ± 3.1	**<0.01**
**Global longitudinal strain (%)**		−22.8 ± 3.9	−22.9 ± 2.8	−24.8 ± 3.0	−25.0 ± 3.4	0.23
**Global circumferential strain rate (s^−1^)**	systole	−1.5 ± 0.3	−1.8 ± 0.4	−1.9 ± 0.6	−1.8 ± 0.2	0.17
	diastole	1.7 ± 0.4	2.1 ± 0.6	2.4 ± 0.6	2.3 ± 0.4	**<0.01**
**Global longitudinal strain rate (s^−1^)**	systole	−1.3 ± 0.5	−1.2 ± 0.2	−1.4 ± 0.6	−1.3 ± 0.3	0.95
	diastole	1.5 ± 0.4	1.4 ± 0.4	1.7 ± 0.4	1.7 ± 0.3	0.31

Continuous variables represented as mean ± standard deviation. Bolded values indicate statistically significant difference (*p*-value < 0.05).

**Table 3 children-10-00271-t003:** Myocardial deformation data for patients and controls.

Patient Data v. Control Data		Controls (n = 18)	T2* < 10 ms (n = 9)	T2* 11–20 ms (n = 10)	T2* 21–30 ms (n = 8)	T2* > 30 ms (n = 22)
			Pair-Wise Comparison to Controls (*p* Values)			
Global circumferential strain (%)		**−31.4 ± 3.9**	**<0.013**	**0.52**	**0.27**	0.05
Global longitudinal strain (%)		−26.2 ± 3.6	0.04	0.02	0.37	0.29
Global circumferential strain rate (s^−1^)	systole	−1.5 ± 0.3	0.82	0.02	0.16	**<0.013**
	diastole	1.5 ± 0.2	0.11	**<0.013**	**<0.013**	**<0.013**
Global longitudinal strain rate (s^−1^)	systole	−1.2 ± 0.3	0.49	0.12	0.10	**<0.013**
	diastole	1.1 ± 0.2	**<0.013**	**<0.013**	**<0.013**	**<0.013**

Continuous variables represented as mean ± standard deviation. Bolded values indicate statistically significant difference with *p*-value < 0.013 with the Bonferroni correction.

**Table 4 children-10-00271-t004:** Correlations between hematologic variables and T2* groups represented by Pearson’s correlation r (*p*-value).

		T2* < 10 ms (n = 9)	T2* 11–20 ms (n = 10)	T2* 21–30 ms (n = 8)	T2* >30 ms (n = 22)	All T2* Groups
Transfusion volume						
Global circumferential strain (%)		0.48 (0.31)	−0.47 (0.28)	−0.14 (0.56)	−0.27 (0.37)	−0.12 (>0.64)
Global longitudinal strain (%)		0.35 (0.30)	**−0.76 (0.03)**	−0.37 (0.38)	−0.24 (0.53)	−0.15 (0.68)
Global circumferential strain rate (s^−1^)	systole	0.35 (0.39)	**−0.73 (0.04)**	−0.21 (0.69)	−0.14 (0.68)	−0.22 (0.22)
	diastole	0.42 (0.29)	−0.59 (0.12)	0.01 (0.98)	−0.43 (0.19)	−0.20 (0.26)
Global longitudinal strain rate (s^−1^)	systole	0.25 (0.53)	**−0.72 (0.04)**	0.20 (0.31)	−0.51 (0.11)	−0.24 (0.18)
	diastole	0.29 (0.49)	**−0.89 (<0.01)**	0.49 (0.31)	−0.35 (0.28)	−0.16 (0.37)
**Ferritin**						
Global circumferential strain (%)		0.41 (0.31)	0.41 (0.81)	−0.07 (0.81)	−0.09 (0.73)	**−0.38 (0.01)**
Global longitudinal strain (%)		0.15 (0.62)	0.15 (0.65)	0.07 (0.84)	−0.2 (0.64)	−0.12 (0.58)
Global circumferential strain rate (s^−1^)	systole	0.22 (0.50)	0.22 (0.57)	0.01 (0.98)	0.27 (0.31)	−0.19 (0.21)
	diastole	0.17 (0.65)	0.18 (0.65)	0.36 (0.38)	0.32 (0.22)	−0.24 (0.12)
Global longitudinal strain rate (s^−1^)	systole	−0.03 (0.92)	−0.03 (0.92)	0.23 (0.58)	0.14 (0.60)	−0.03 (0.82)
	diastole	−0.21 (0.56)	−0.21 (0.59)	0.46 (0.26)	0.25 (0.36)	−0.14 (0.37)
**Hemoglobin**						
Global circumferential strain (%)		−0.51 (0.18)	0.51 (0.14)	0.51 (0.29)	**0.53 (0.03)**	−0.02 (0.85)
Global longitudinal strain (%)		−0.38 (0.33)	0.60 (0.08)	0.60 (0.11)	−0.25 (0.36)	−0.12 (0.47)
Global circumferential strain rate (s^−1^)	systole	0.33 (0.34)	0.33 (0.34)	−0.46 (0.24)	−0.09 (0.70)	−0.16 (0.29)
	diastole	0.24 (0.49)	0.24 (0.50)	0.07 (0.86)	−0.13 (0.60)	−0.15 (0.33)
Global longitudinal strain rate (s^−1^)	systole	0.28 (0.42)	0.28 (0.42)	−0.13 (0.76))	−0.40 (0.11)	−0.10 (0.52)
	diastole	0.22 (0.52)	0.22 (0.52)	0.35 (0.40)	**−0.50 (0.04)**	−0.10 (0.48)

Bolded values indicate statistically significant difference (*p*-value < 0.05).

## Data Availability

The data presented in this study are available on request from the corresponding author. The data are not publicly available due to IRB restrictions.

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
