# Peer review of "Cardiac Magnetic Resonance Strain in Beta Thalassemia Major Correlates with Cardiac Iron Overload"

_children, 2023, doi:10.3390/children10020271_

Round 1

Reviewer 1 Report

I will like to congratulate the authors for their  work. The correlation between GCS and T2* while not surprising is interesting. However, as appropriately mentioned in the discussions, the relevance of findings are significantly impacted by limitations. Nevertheless, its an interesting correlation and this paper will provide a good stepping stone for future research.

Author Response

Thank you for the reviewer's comments.  We look forward to continuing to explore further utility of CMR strain in this patient population.

Reviewer 2 Report

I think that the article is very interesting and well written.

I only suggest the following technical note. In the Discussion section, lines 263-266, i suggest to add the following sentence: “Our negative finding in reference to longitudinal strain may be secondary to some of the inherent pitfalls with feature tracking at the basal segments due to the thinner wall particularly at the hinge points of the mitral valve annulus [10] and/or due to the possible extrinsic sternal compression on basal segments which is more enhanced in individuals with anterior chest wall deformity, such as those with pectus excavatum”. Please add the following reference: Sonaglioni A, Nicolosi GL, Rigamonti E, Lombardo M, La Sala L. Molecular Approaches and Echocardiographic Deformation Imaging in Detecting Myocardial Fibrosis. Int J Mol Sci. 2022 Sep 19;23(18):10944. doi: 10.3390/ijms231810944. PMID: 36142856; PMCID: PMC9501415.

Author Response

The authors are grateful for the reviewer's comments. The recommended sentence and reference has been added to the manuscript and is represented in lines 268-270.